# Direct observational evidence from space of the effect of $CO_2$ increase on longwave spectral radiances: the unique role of high-spectral-resolution measurements

**João Teixeira, Robert C. Wilson, and Heidar T. Thrastarson**

Jet Propulsion Laboratory, California Institute of Technology, Pasadena, California 91109, USA

**Correspondence:** João Teixeira (teixeira@jpl.nasa.gov)

**Abstract.** TS1 TS2 CE1 We present a direct measurement of the impact of increased atmospheric $CO_2$ on the spectra of the Earth's longwave radiation obtained from space. The goal of this study is to experimentally confirm that the direct effects of $CO_2$ increase on the Earth's outgoing longwave spectra follow theoretical estimates, by developing a methodology that allows for a direct and more precise comparison between theory and observations. In this methodology, a search is performed to find selected ensembles of observed atmospheric vertical profiles of temperature and water vapor that are as close as possible to each other in terms of their values. By analyzing the spectral radiances measured from space by the Atmospheric Infrared Sounder (AIRS), corresponding to the selected ensembles of profiles, the effects of increased $CO_2$ on the spectra can be isolated from the temperature and water vapor effects. The results illustrate the impact of the increase in $CO_2$ on the longwave spectra and compare well with theoretical estimates. As far as the authors are aware, this is the first time that the spectral signature of the increase in $CO_2$ (isolated from temperature and water vapor changes) has been directly observed from space.

## 1 Introduction

As is clearly discussed in the excellent historical compilation of Archer and Pierrehumbert (2011), as well as the essential references therein, it has been known for decades that an increase in atmospheric greenhouse gases such as $CO_2$ can lead to global warming essentially by changing the longwave radiative fluxes of energy at the top of the atmosphere. While remarkable progress has been made over the last few decades in laboratory, theoretical, modeling, and prediction studies of the physics of climate change (e.g., Plass, 1956; Manabe and Wetherald, 1967, 1975; Hansen et al., 1984; Ramanathan, 1988; Ramaswamy et al., 2018), experimental confirmation from space of the direct effects of $CO_2$ (independent from temperature and water vapor changes) on the Earth's outgoing longwave spectra has been elusive. The goal of the present study is to experimentally confirm that the direct effects of $CO_2$ increase on the Earth's outgoing longwave spectra follow theoretical estimates, by developing a methodology that allows for a direct and more precise comparison between theory and observations.

Kiehl (1983) discussed the possibility of utilizing spectrally resolved satellite measurements of longwave radiation to detect and characterize the impact of climate change on the longwave spectra and simulated the changes in clear-sky spectra due to increases in $CO_2$ and temperature. This pioneering modeling study has been followed by other modeling studies focused on the spectral signature of climate change (e.g., Mlynczak et al., 2016; Brindley and Bantges, 2016).

The fact that measurements from space of the direct effects of increased $CO_2$ on the longwave spectra have been notoriously difficult to obtain is associated with the sparsity of high-spectral-resolution observations of longwave radiances before the early 21st century and with the challenge of disentangling the effects of $CO_2$, temperature, and water vapor on

the spectral radiances. While measurements of the spectral effects of the combined changes in $CO_2$, temperature, water vapor, and other gases have been published (e.g., Harries et al., 2001; Brindley and Bantges, 2016; Strow and DeSouza-Machado, 2020; Whitburn et al., 2021; Huang et al., 2022; Raghuraman et al., 2023), the direct effects of $CO_2$ alone have been difficult to depict accurately.

For example, in a pioneering study, Harries et al. (2001) calculate the spectral differences between two infrared instruments, one launched in the 1970s and the other in the 1990s. Despite the difficulties of accurately estimating spectral differences between two different instruments, Harries et al. (2001) are able to discern and potentially assign, using model simulations, some of the spectral differences to changes in greenhouse gases such as $CO_2$. However, they do not attempt to disentangle the effects due purely to $CO_2$ from temperature and water vapor changes in the observational data. In a recent study, De Longueville et al. (2021) illustrate the increased $CO_2$ absorption in the Infrared Atmospheric Sounding Interferometer (IASI) spectra from 2008 to 2017, although they do not isolate it from the joint effects of temperature and water vapor.

The recent studies of Strow and DeSouza-Machado (2020) and Huang et al. (2022) investigate in detail the Atmospheric Infrared Sounder (AIRS) instrument radiance trends over the last several years and highlight the remarkable stability of the AIRS radiance record while also discussing the role of temperature, water vapor, $CO_2$, and other gases in framing the nature of the AIRS radiances. However, they only isolate these effects using modeling/theoretical approaches. These studies do not attempt to disentangle the impact of $CO_2$ (or other gases) from temperature and water vapor in the observational data directly.

In the present study, a new methodology is proposed for a more direct measurement of the effect of $CO_2$ increase on longwave spectral radiances in such a way as to provide a direct and more precise comparison to theoretically derived radiances. The goal of this approach is to isolate the effects of $CO_2$ from the effects of temperature and water vapor. This is achieved by searching for atmospheric profiles of temperature and water vapor that are as close to each other as possible (referred to as analogues) but that have $CO_2$ concentrations that are significantly different. Measuring from space the spectral radiances that correspond to these analogues allows us to detect, given the right circumstances, the unique impact of $CO_2$ on the radiances with enough precision and accuracy in key spectral regions. Specifically, in this work, 1000 temperature and water vapor reference profiles are selected and a search is performed for analogue profiles that are close to the reference profiles to within a specified uncertainty range.

## 2   Observational data

The spectral longwave radiances are measured by the Atmospheric Infrared Sounder (AIRS), and the profiles of atmospheric temperature and water vapor, as well as cloud properties, are from retrievals that include data from the AIRS and AMSU (Advanced Microwave Sounding Unit) suite of instruments on Aqua (e.g., Aumann et al., 2003; Chahine et al., 2006) as well as other datasets (see Appendix A TS3).

## 3   Results

A set of 1000 temperature and water vapor reference profiles from July 2003 are randomly selected, albeit obeying the following constraints: these profiles are over the tropical/subtropical oceans (30° S to 30° N) TS4, with cloud cover less than 10 % and within a sea surface temperature (SST) range of 298 to 302 K.

For each of these 1000 reference profiles, a search is performed to find analogue profiles that are within absolute value thresholds of 1.4 K for temperature and $1.4\,\mathrm{g\,kg^{-1}}$ for water vapor at any vertical level (with respect to the corresponding reference profiles). The search spans a period from 2003 to 2012 but only includes June–July–August (JJA) for each year. These analogue profiles are also over the tropical/subtropical oceans and in (almost) clear sky (cloud cover less than 10 %), and the analogue SST differences are also within 1.4 K. For each of these analogue profiles, the corresponding cloud-cleared (e.g., Susskind et al., 2003) AIRS spectral radiances are selected.

To estimate the impact of $CO_2$ increase on the observed spectral radiances, the differences between the radiances observed at the location and time of each analogue and the radiances observed at the location and time of the corresponding reference profile are calculated. These differences (which correspond to different reference profiles and different years from 2003 to 2012) are aggregated to provide an estimate of the overall annual mean difference. These differences between the spectral radiances – which are measured at different years and as such reflect different amounts of $CO_2$ – are compared with theoretical estimates of the radiance impact of $CO_2$. A key assumption (which is discussed below) is that the annual mean spectral radiance differences corresponding to each reference state are (to first order) not sensitive to the reference state itself for these selected reference profiles.

To estimate the theoretical values, the spectral radiances corresponding to the reference temperature and water vapor profiles are simulated with different values of $CO_2$ concentration that reflect its mean increase from 2003 to 2012 as measured by the National Oceanic and Atmospheric Administration (NOAA) Mauna Loa station (see Appendix A). The kCARTA forward model (Strow et al., 1998; DeSouza-Machado et al., 2020) is used to simulate the spectral radiances and is convolved with the AIRS spectral response functions to obtain theoretical AIRS radiances.

To investigate the uncertainties associated with the temperature and water vapor thresholds used to search for analogues, a preliminary theoretical study of these uncertainties is performed: using one pair (temperature and water vapor) of reference profiles, 1000 synthetic temperature and water vapor analogues are created by drawing from a normal distribution defined by a zero mean and standard deviations of 0.5 K and 0.5 g kg$^{-1}$ (which are close to the values estimated based on the observed temperature and water vapor analogues corresponding to the 1000 observed reference profiles) with the constraint that the absolute difference values at any level cannot be larger than the thresholds of 1.4 K and 1.4 g kg$^{-1}$. Theoretical spectral radiances are calculated for each of these 1000 synthetic analogue pairs of profiles, and these theoretical values are used to estimate the impact of the temperature and water vapor thresholds on the spectral radiances.

Figure 1 compares the theoretical spectral radiance differences due to the annual mean increase in $CO_2$ (for that reference profile) during this period with the mean spectral differences between the 1000 (synthetic) analogues and the reference profile. Three specific lines are shown in this context: one that includes all the 1000 synthetic spectral radiances and two additional ones in which radiance difference outliers which are larger than 0.5 or 1 mW m$^{-2}$ sr$^{-1}$ (cm$^{-1}$)$^{-1}$ are filtered out. The figure is focused on the 680-to-780 cm$^{-1}$ spectral range where the $CO_2$ effect is more prominent (see below).

In the spectral region between 680 and 720 cm$^{-1}$, the spectral radiance differences due to temperature and water vapor uncertainties are much smaller than the spectral radiance differences due solely to $CO_2$ increase, for all the analogue difference lines. Between 720 and around 750 cm$^{-1}$, although the analogue lines are starting to diverge among themselves in certain regions, their values are still smaller than the $CO_2$ differences. For values above around 750 cm$^{-1}$ where the $CO_2$ impact is reduced while the impact of water vapor becomes dominant, the analogue difference lines are of the same order of magnitude as the $CO_2$ difference lines – although the analogue difference values that are calculated using the filter value of 0.5 mW m$^{-2}$ sr$^{-1}$ (cm$^{-1}$)$^{-1}$ are in certain spectral regions clearly smaller than the $CO_2$ differences. Given the apparent small impact of these temperature and water vapor uncertainties on the spectral radiances in certain key spectral regions, these results provide a degree of confidence in the methodology.

Figure 2 shows the annual mean differences in terms of spectral radiances due to $CO_2$ increase for the AIRS observations and the theoretical values, together with the standard deviation of the observations. This figure is focused on the 680-to-780 cm$^{-1}$ spectral range, which represents the R branch of the 15 μm $CO_2$ band and is a spectral region where the $CO_2$ signal is particularly significant. In this spectral region, the enhanced absorption within the troposphere, where temperature decreases with height, leads to a reduction in the outgoing radiation. From a broader climate perspective, the reduction in outgoing radiation is behind the increase in global surface temperature that is necessary for the overall climate system to re-establish energy balance at the top of the atmosphere, and as such it is a critical component of global warming. During this period, the measured monthly mean $CO_2$ mole fraction at Mauna Loa increased on average by approximately 2 ppm yr$^{-1}$ [TS6] (see Appendix A).

The theoretical annual mean differences and associated standard deviations are calculated based on the 1000 reference profiles. These standard deviations are not shown because they are so small that they would be almost imperceptible in the figure. This apparent lack of theoretical sensitivity to the reference states supports the key assumption mentioned above that the annual mean spectral radiance differences corresponding to each reference state are (to first order) not particularly sensitive to the reference state within this specific set of reference profiles.

Figure 2 illustrates that the new methodology leads to observed radiance differences that are close to the theory. Despite some noise and a small negative bias in some spectral regions, there is good agreement between theory and observations, with the observations following closely the theoretical impact of increased $CO_2$. It is noticeable that even in the 750–780 cm$^{-1}$ spectral region, where water vapor plays a large role and larger uncertainties are expected based on Fig. 1 and the standard deviations, the observations in Fig. 2 match the theory to a good level of accuracy.

The sensitivity of the observations to different aspects has been analyzed. In particular, the observations shown in Fig. 2 correspond to observed scan angles between −5 and 5° from nadir (see Appendix A), and the theoretical radiances are estimated at nadir. To remove outliers and to select analogues that are as close as possible to the reference states, analogues that have absolute radiance differences, as compared to the reference states, that are larger than 0.5 mW m$^{-2}$ sr$^{-1}$ (cm$^{-1}$)$^{-1}$ are filtered out. Overall, the observed spectral radiances shown in Fig. 2 correspond to about 300 analogues. Results obtained with scan angles between −10 and 10° and with outlier filter values of 1 mW m$^{-2}$ sr$^{-1}$ (cm$^{-1}$)$^{-1}$ show no meaningful differences, highlighting the robustness of the methodology.

Given the spatial and temporal variability in $CO_2$, the lack of accurate knowledge of the $CO_2$ values for each specific reference state and corresponding analogues leads to uncertainties in the theoretical radiance estimates. A preliminary analysis of this uncertainty (not shown) appears to indicate that the observations, in some regions where there are biases, could still be within the theoretical standard deviation range if the $CO_2$ uncertainties were to be considered explicitly. In fact, using data from the Orbiting Carbon Observatory-2 (Crisp et al., 2004), this analysis leads to theoretical standard deviations of the order of 0.025 mW m$^{-2}$ sr$^{-1}$ (cm$^{-1}$)$^{-1}$ in several channels in the region between 700 and 740 cm$^{-1}$. This suggests that some of the observational biases could be, at least partly, related to $CO_2$ uncertainties. Small biases

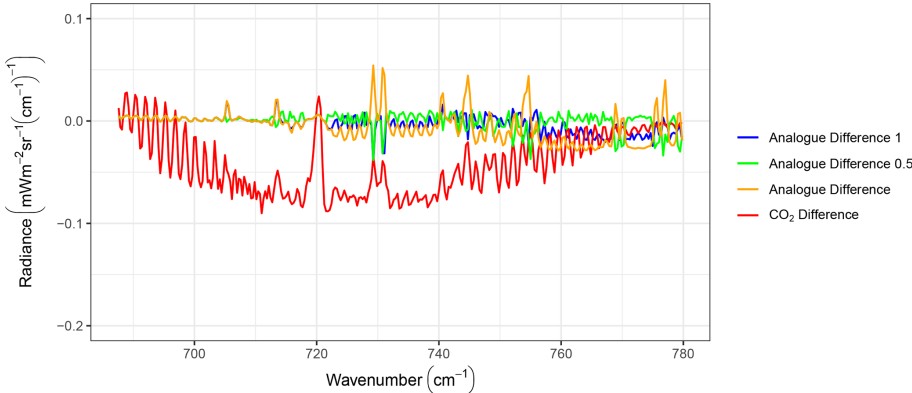

**Figure 1.** Theoretical spectral radiance differences (in mW m$^{-2}$ sr$^{-1}$ (cm$^{-1}$)$^{-1}$) due to the annual mean increase in CO$_2$ during the period from 2003 to 2012 (red line) together with the theoretical mean radiance differences between the 1000 synthetic temperature and water vapor profiles and the reference profile. Three specific lines are shown: one that includes all the 1000 synthetic radiances (orange line) and two additional lines in which radiance difference outliers which are larger than 0.5 mW m$^{-2}$ sr$^{-1}$ (cm$^{-1}$)$^{-1}$ (green line) or 1 mW m$^{-2}$ sr$^{-1}$ (cm$^{-1}$)$^{-1}$ (blue line) are filtered out.

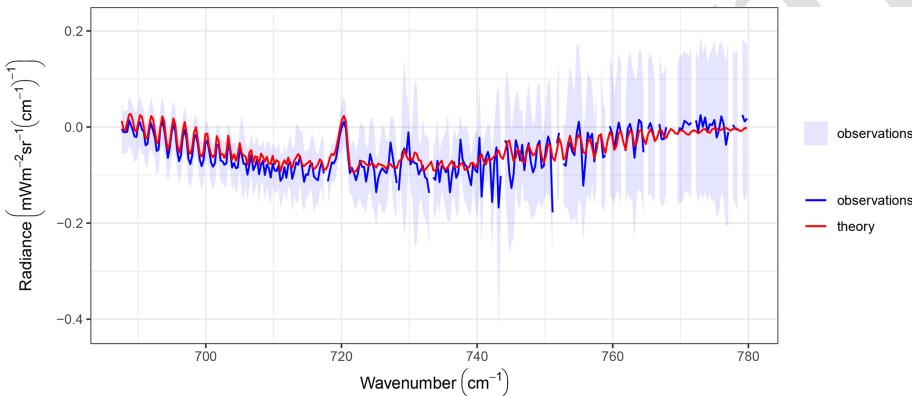

**Figure 2.** Annual mean radiance differences (in mW m$^{-2}$ sr$^{-1}$ (cm$^{-1}$)$^{-1}$) due to CO$_2$ increase from the AIRS observations (blue line) and from theory (red line) and standard deviations for the AIRS observations (blue shading), following the methodology described and illustrating the direct impact of CO$_2$ increase on the spectral radiances during the 2003–2012 period (see text for details). Based on observations with scan angles between −5 and 5°.

of the radiative transfer model could also potentially be behind some of the differences. A more detailed uncertainty study focused on these two critical aspects would be a natural follow-up to this work.

## 4 Conclusions

We present a direct measurement of the impact of increased atmospheric CO$_2$ on the spectra of the Earth's longwave radiation obtained from space. The approach involves a new methodology to disentangle the impact of CO$_2$ on the observed longwave spectral radiances from the effects of temperature and water vapor in such a way as to provide a direct and more precise comparison with theoretical estimates of the radiance impact of CO$_2$. The observations obtained using this methodology compare well with theoretical esti-

mates of the direct CO$_2$ radiative impact on the Earth's longwave spectra.

In the future, variants of this methodology could be used to isolate the observational radiative impact of different physical and chemical properties of the climate system and as such provide a better observational depiction of the Earth's radiative forcing (e.g., Huang et al., 2016; Ramaswamy et al., 2018) and of climate feedbacks.

The stability of the AIRS instrument has been determined to be about 1 order of magnitude smaller (better) than the climate temporal signal for this spectral region (Strow and DeSouza-Machado, 2020), providing a high level of confidence in the results presented. This work also illustrates the unique and critical role of accurate and stable hyperspectral infrared observations from space in addressing fundamental climate physics questions.

This new methodology can undoubtedly be refined and its uncertainties better characterized and understood to establish its accuracy and precision more clearly. But as far as the authors are aware, this study represents the first attempt to establish a more precise experimental confirmation from space of the direct effects of $CO_2$ on longwave spectral radiances. The results (solely based on observations) confirm that the effects of the recent atmospheric $CO_2$ increase on longwave spectral radiances follow theoretical estimates. As such, these results confirm a critical foundation of the science of global warming.

## Appendix A: Data and methods

In this work, the focus is on sets of temperature and water vapor profiles (and the corresponding spectral radiances) that belong to a common physical regime associated with clear-sky (or with negligible cloud amounts) atmospheric thermodynamics over the tropical and subtropical oceans (from $30°$ S to $30°$ N). Figures A1 and A2 compare the mean temperature and water vapor reference profiles with the JJA climatological mean profiles from the AIRS–AMSU retrieval for this regime and illustrate that the reference profiles are indeed representative of the atmospheric thermodynamic climatology of this regime.

AIRS is a hyperspectral sounder on the Aqua spacecraft (Parkinson, 2003) covering the $3.7$–$15.4\,\mu$m infrared spectral region with 2378 channels and a spatial resolution of $13.5$ km at nadir. AIRS was launched into a 705 km altitude orbit on 4 May 2002 and has been in routine data gathering mode essentially uninterrupted since September 2002. The 13:30 LT TS7 ascending node and orbital altitude of the Aqua spacecraft orbit have been accurately maintained (until 2022), and daily (nearly) global coverage is essentially achieved from the ascending and descending orbits. The AIRS radiometric accuracy has been discussed in several studies (e.g., Pagano et al., 2003; Tobin et al., 2006; Aumann et al., 2006). Detailed prelaunch radiometric calibration tests showed that the AIRS radiometric calibration was accurate to within $0.2$ K for scene temperatures between 205 and 310 K (Pagano et al., 2003). As an example, they show that the residual radiometric accuracy compared to an external blackbody at 265 K is between $-0.2$ and $0.1$ K for the vast majority of AIRS channels, and in particular it is between $-0.2$ and $0$ K in the 15 μm spectral region.

AIRS radiances are routinely assimilated in all major global weather prediction systems and are used to retrieve vertical profiles of atmospheric temperature, water vapor, and key atmospheric constituents as well as cloud and surface parameters (e.g., Susskind et al., 2003; Smith and Barnet, 2020).

Recently, Strow and DeSouza-Machado (2020) confirmed that the AIRS instrument stability for about 400 channels is within $2 \times 10^{-3}\,\mathrm{K\,yr^{-1}}$ in brightness temperature, which is about 1 order of magnitude smaller than the climate temporal signal in brightness temperature for the spectral region that is investigated in this study. Note that, according to Huang et al. (2022), the trend due to the AIRS instrument spectral shift is also within $2 \times 10^{-3}\,\mathrm{K\,yr^{-1}}$.

The Advanced Microwave Sounding Unit (AMSU) on Aqua is a 15-channel microwave (MW) instrument with 12 temperature sounding channels in the 50–58 GHz oxygen absorption band that are used to produce an AMSU MW-only retrieved temperature profile dataset (Rosenkranz, 2001) and are also part of the AIRS–AMSU retrieval.

The atmospheric profiles of temperature and water vapor used in this study are from the AIRS–AMSU retrieval products that are based on level-1 data from AIRS and AMSU, as well as on a neural network retrieval first guess (Milstein and Blackwell, 2016) that uses the European Centre for Medium-Range Weather Forecasts (ECMWF) analyses as the training dataset. In this context, the AIRS–AMSU retrieved profiles of atmospheric temperature and water vapor depend directly on AIRS and AMSU data and indirectly on the ECMWF data-assimilation system as well as on a variety of observational datasets that are assimilated by ECMWF (e.g., radiosondes, global navigation satellite system (GNSS) radio occultation (RO), other infrared (IR) and MW sounders) via a neural network retrieval algorithm. Specifically, version 6 of the AIRS–AMSU retrieval products is used. The standard pressure levels for the retrieved temperature and water vapor profiles are described in https://docserver.gesdisc.eosdis.nasa.gov/public/project/AIRS/V7_L2_Standard_Pressure_Levels.pdf (last access: TS8).

For temperature, reference profiles and their respective analogues are extracted for both the AIRS–AMSU and the AMSU MW-only products. Although the AIRS–AMSU retrievals utilize a large variety of different sources of information about the atmosphere (as described above), the AMSU MW-only retrievals (which are based on the oxygen band and have no dependency on $CO_2$) are used here as a (somewhat) independent temperature profile dataset to partly validate the methodology.

Figure A1 shows the mean temperature reference profiles for AIRS–AMSU and MW-only retrievals, together with the mean AIRS–AMSU temperature climatology profile for this regime (i.e., the mean of all profiles from JJA 2003 following the constraints referred to above that define this regime). Also shown are the standard deviations of all the analogue profile (from both AIRS–AMSU and MW-only) differences versus the corresponding reference profiles. As mentioned above, this figure illustrates that the reference profiles are representative of this atmospheric thermodynamic regime (as characterized by the AIRS–AMSU climatology). In addition, it shows that the AIRS–AMSU and MW-only reference profiles and their analogues are close to each other. The fact that the analogues have similar characteristics in both the AIRS–AMSU (which uses a variety of sources of information) and

the AMSU MW-only retrievals (which are based on the oxygen band) suggests that the analogues appear independent of specific $CO_2$ assumptions in the retrieval algorithms and in the ECMWF data-assimilation system. Figure A2 shows similar results for the water vapor profiles.

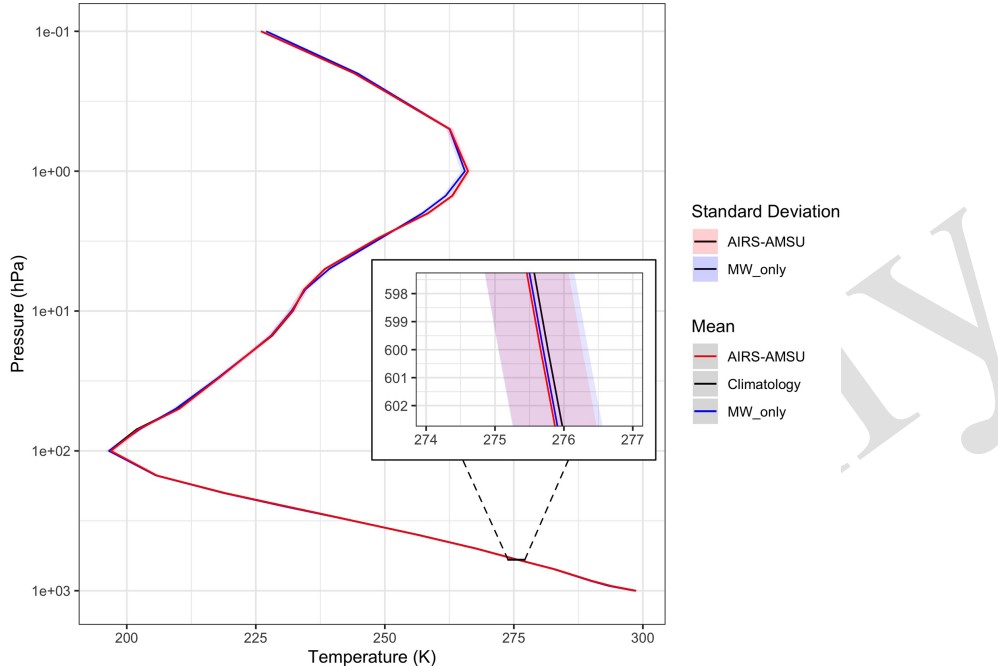

**Figure A1.** Mean temperature (in K) reference profiles for AIRS–AMSU (red line) and MW-only (blue line) retrievals, together with the mean climatology AIRS–AMSU profile (black line) for this regime. Also shown are the profiles of the standard deviations of all the analogue differences (versus the reference profiles) from both the AIRS–AMSU (red shading) and the MW-only (blue shading) analogues.

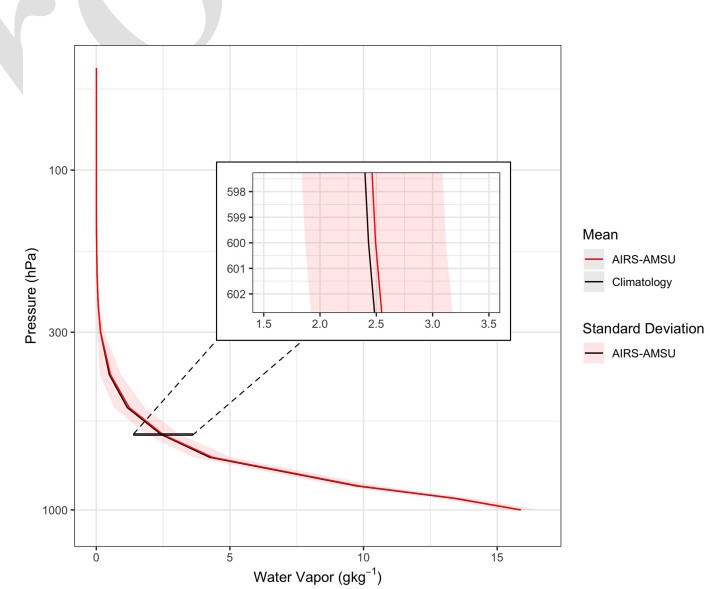

**Figure A2.** Mean water vapor (in g kg$^{-1}$) reference profiles for the AIRS–AMSU (red line) and the mean climatology AIRS–AMSU profile (black line) for this regime. Also shown is the profile of the standard deviations of all the analogue differences (versus the reference profiles) from the AIRS–AMSU (red shading) analogues.

Note that the temperature and water vapor profile retrieval products discussed above are only used to find analogues and that, in this context, other observational estimates of temperature and water vapor could be used for this purpose, e.g., from other IR sounder retrievals such as IASI or the Cross-track Infrared Sounder (CrIS), from other microwave sounder retrievals, or from analysis and re-analysis products. The key reason why AIRS–AMSU temperature and water vapor profiles are used to implement this analogue methodology is their inherent collocation with the AIRS radiance spectra which are directly used in this study.

Smaller threshold values lead to analogues that are closer to the corresponding reference profiles but to a smaller number of analogues, while larger threshold values lead to analogues that are farther from the corresponding reference profiles but to a larger number of analogues. Based on our initial studies, the specific thresholds of 1.4 K and 1.4 g kg$^{-1}$ that are used here are a compromise between these extremes. For example, using the same methodology for 1000 reference profiles and thresholds of 1 K and 1 g kg$^{-1}$ does not yield a sufficient number of analogues for precise results to be obtained.

Given the steady increase in global $CO_2$, the best way to make sure that analogues with significantly different values of $CO_2$ are selected is to search for these analogues over several years – 10 years from 2003 to 2012 in our study.

The theoretical spectral radiances are calculated for different values of $CO_2$ that correspond to the mean July $CO_2$ values for 2003 to 2012, as measured by NOAA at Mauna Loa (e.g., Thoning et al., 1989). Note that uncertainty associated with the lack of accurate knowledge of $CO_2$ values for each analogue profile, due to spatial and temporal variability, can lead to noteworthy uncertainties in the theoretical spectral radiances.

Following Huang et al. (2022), to simplify the implementation of the methodology and to circumvent the influence of different viewing zenith angles on the overall results, only spectra measured within scan angles between $-5$ and $5°$ from nadir are presented – although spectra measured within scan angles between $-10$ and $10°$ from nadir are also analyzed to evaluate the sensitivity to scan angle. For consistency, the theoretical radiances are calculated at nadir. Future studies to evaluate the impact of different viewing angles on the results and to increase the observational sample will be performed.

In Fig. 2, the observational standard deviations grow from values of the order of 0.01 mW m$^{-2}$ sr$^{-1}$ (cm$^{-1}$)$^{-1}$ close to 700 cm$^{-1}$ to values larger than 0.1 mW m$^{-2}$ sr$^{-1}$ (cm$^{-1}$)$^{-1}$ close to 780 cm$^{-1}$.

For perspective, when analyzing Fig. 2, note that changes in temperature and water vapor of the order of the ones experienced by the atmosphere during the first 20 years of the 21st century lead to positive theoretical changes in brightness temperature that are fairly constant in this spectral interval and that correspond to approximately 25 % (in absolute values) of the $CO_2$ theoretical changes in the 720–740 cm$^{-1}$ region, increasing (in percentage) to much higher values toward both the 680 and the 780 cm$^{-1}$ extremes of the figure (e.g., Huang et al., 2022).

These results provide evidence that (i) AIRS has the stability required to address, in an accurate and precise manner, climate change questions of the nature described here and (ii) space-based spectral measurements are becoming of comparable quality to prior spectroscopic estimates. Note that Strow and DeSouza-Machado (2020) estimate that the trend uncertainty due to $CO_2$ spectroscopy uncertainties is of the order of their estimate for the stability of the AIRS instrument in these channels at around $2 \times 10^{-3}$ K yr$^{-1}$. The similarity between the observations and theory in Fig. 2 further supports this point.

The results presented are for clear sky over the tropical and subtropical oceans. Other regions of the globe and physical regimes will be addressed in the future. Performing a similar study that includes cloud effects to address the all-sky impact requires overcoming more challenging obstacles both from the observational (analogue) perspective and from the theoretical (radiative forward model) perspective.

**Code availability.** The kCARTA model is available at https://github.com/sergio66/kcarta_gen TS9 (SRCv1.18 was used).

**Data availability.** AIRS–AMSU version 6 and the AMSU MW-only L2 standard products are used for the temperature and water vapor profiles: https://doi.org/10.5067/Aqua/AIRS/DATA201 (AIRS Science Team and Teixeira, 2013) TS10. For AIRS L1B infrared radiances (version 5), see https://doi.org/10.5067/YZEXEVN4JGGJ (AIRS project, 2007) TS11.

**Author contributions.** JT developed the methodology, performed the analysis, and wrote the manuscript. RCW and HTT implemented the methodology and contributed to the analysis and the manuscript.

**Competing interests.** The contact author has declared that none of the authors has any competing interests.

**Disclaimer.** Publisher's note: Copernicus Publications remains neutral with regard to jurisdictional claims made in the text, published maps, institutional affiliations, or any other geographical representation in this paper. While Copernicus Publications makes every effort to include appropriate place names, the final responsibility lies with the authors.

**Acknowledgements.** The authors would like to thank everyone who has been involved in creating the AIRS radiance record. Exciting conversations over the years with several colleagues, including

TS12 L. Strow, C. Barnet, H. Aumann, S. DeSouza-Machado, and X. Huang, and the advice from J. V. C. Teixeira on technical aspects are gratefully acknowledged. The authors would also like to thank the two referees for insightful and valuable reviews.

**Financial support.** This research has been supported by the Jet Propulsion Laboratory, California Institute of Technology, under a contract with the National Aeronautics and Space Administration (grant no. 80NM0018D0004).

**Review statement.** This paper was edited by Andreas Hofzumahaus and Gabriele Stiller and reviewed by Zhao-Cheng Zeng and one anonymous referee.

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

## Remarks from the language copy-editor

## Remarks from the typesetter

**TS9** Please clarify whether the data set is your own. If yes, please provide a DOI in addition to your GitHub URL since our reference standard includes DOIs rather than URLs. If you have not yet created a DOI for your data set, please issue a Zenodo DOI (https://help.github.com/en/github/creating-cloning-and-archiving-repositories/referencing-and-citing-content). If the data set is not your own, please inform us accordingly. In any case, please ensure that you include a reference list entry corresponding to the data set including creators, title, and date of last access.