# Peer review of "Direct Observational Evidence from Space of the Effect of CO2 Increase on Longwave Spectral Radiances: The Unique Role of High Spectral Resolution Measurements"

_EGUsphere, 2023_

## Referee Comment (RC2)

Review of Teixeira et al. "Direct Observational Evidence from Space of the Effect of CO2 Increase on Longwave Spectral Radiances: The Unique Role of High Spectral Resolution Measurements"

This study used AIRS high spectral infrared radiance data to quantify the impact of increasing atmospheric CO2 concentration on the absorption of 15-um CO2 band after eliminating interference from water vapor and temperature changes. They found that the isolated signal from increased absorption by CO2 from AIRS is consistent with RT model simulations. Overall, this letter was well written. The message the authors trying to deliver is clear. The results from this study based on data driven analysis are important to confirm our understanding of CO2 greenhouse effect. However, given its short length, some important details are not there in the paper and therefore lead to confusions. I will list them below.

(1) Some parameters in Experiments A and B are not the same. For example, you analyzed 2005-2015 for A but 2003 to 2012 for B. Is there a reason that the two experiments need to start from different years? Also, you explain the adoption of 1.2k and 1.2g/kg in the appendix for A, but in B, you used 1.4k and 1.4 g/kg. You need to justify these numbers.

(2) In the experiments, you selected one set of profiles for A and 100 sets for B. It is not clear how the set of profiles were selected. How do you make sure they are representative of the temperature and h2o vertical distributions on Earth? It would be better if you can show some of the profiles in the appendix as well.

(3) "In experiment B, a key assumption is that the annual mean spectral radiance differences corresponding to each reference state are (to first order) not sensitive to the reference state itself for these selected reference profiles."
You now have 100 sets of profiles and the corresponding spectral change. Can you use these results to justify your assumptions here? For example, are the temperature (or h2o) variabilities correlated with the spectral radiance differences?

(4) In Figure 2, how large is the uncertainty for the observations? You have that for Figure 1 but not Figure 2. Also, the mismatch between observations and theoretical calculations are large over those CO2 absorption line centers. The difference can be 0.04K for the lines on the left of 700 cm-1, which is larger than the expected spectra noise. You attributed this difference to CO2 uncertainty. Can you reconcile the two if you increase the CO2 in your RT model? It seems your current calculations have less absorption over those lines.

Minor comments:
(1) Line 181-182, the absolute radiometric calibration accuracy is usually temperature dependence. For this 0.2K accuracy, is it relative to what temperature blackbody?
(2) Line 206-207, do you have figure or reference to justify that the temperature profiles in AIRS/AMSU and AMSU MW-only are similar?
(3) In Longueville et al. (2021), the authors showed Figure 2 to illustrate the increased CO2 absorption in the IASI spectra from 2008 to 2017, though they did not isolate it from the joint effects of temperature and h2o. This is a related reference for this study.
De Longueville, H., Clarisse, L., Whitburn, S., Franco, B., Bauduin, S., Clerbaux, C., et al. (2021). Identification of short and long-lived atmospheric trace gases from IASI space observations. Geophysical Research Letters, 48, e2020GL091742. https://doi. org/10.1029/2020GL091742

---

## Author Comment (AC1)

**Reply to the review from referee 1**

Note: the original text from the review is in black and our replies are in blue.

While past studies have evaluated and interpreted the effects of CO2 on IR spectral changes from observations, none have done so from a purely observational standpoint. Instead, the past studies have relied on modeling or theoretical interpretation to separate the direct effects of CO2 change from the effects of temperature and water vapor changes that also occur at the relevant CO2 abortion bands. This manuscript represents the first successful attempt to perform that isolation solely using observations. To do so, the authors search for profiles over different years with significantly different CO2 concentrations, but with very similar T and WV profiles. They then quantify the difference between the corresponding spectral radiances (between a reference year and more recent year) to demonstrate that the expected isolated effect of CO2 to reduce OLR is evident in the AIRS observations over the tropospheric CO2 absorption band evaluated in this study. The authors also perform radiative transfer calculations solely with changes in CO2 concentration, as further support that they are truly isolating the effects of CO2 in their observational estimates. Their work will certainly be of interest to ACP Letters readers and marks an important milestone in observing the effects of CO2 on the climate. I provide some minor comments below that will hopefully help improve the manuscript.

We would like to thank the referee for a detailed and constructive review. All the comments and suggestions by the referee have been addressed and have been extremely helpful in improving the manuscript. The revised version will be an improved manuscript because of the referee's comments and suggestions.

To identify analogous profiles, the authors use RMS difference thresholds of 1.2 K for temperature and 1.2 g/kg for water vapor to identify analogues to the reference profiles and thresholds of 1.4 K and g/kg for Experiment B. Some evidence should be provided that those thresholds do indeed, represent sufficiently small radiative effects from T and WV changes. One could suspect a 1.2 K temperature change, even if just locally, could have a significant radiative response relative to the influence of CO2 (for instance, thinking in the context of a climate radiative feedback). One option is to run both the reference and analogue profiles through kCarta, with the same CO2 concentrations, and show the radiative effects from any T and WV are small compared to the direct CO2 effects

We follow the referee's suggestion and in the revised version of the paper we will have a new figure that compares the theoretically expected spectral differences due to changes in CO2 with spectral differences due to temperature and water vapor changes, for the spectral region that we are focused on. This will help quantify the spectral radiance uncertainty due to the uncertainty in temperature and water vapor from the analogues.

Line 70-73: The authors should explain why it is important to stay as close to nadir as possible. Although they explain in the appendix that doing so leads to smaller biases relative to the theoretical calculations, it would be helpful to mention why that is the case.

Text will be added to the revised version to clarify this issue.

It's not clear why the authors chose to publish both experiment A and B. Experiment A seems like a light test of the methodology before performing the more robust Experiment B. I can understand performing A while putting this study together, but its not clear why the authors have chosen to feature the results of A so prominently in the manuscript (and have given it a figure). I suspect the authors have good reason for doing so, but it does not come across clearly in the text. I worry someone who skims this Letter won't realize Figure 2 is the more robust, important figure.

We agree with the referee and in the revised version of the paper we will delete the current figure 1, and while we will still briefly mention some of the key points of the figure, we will not organize the paper in the same way. Basically, we will not divide the paper in experiment A and experiment B sections. We will focus on what we previously referred to as experiment B.

For Figure 1, experiment A, the observed radiance difference has a clear negative bias relative to theoretical for both experiments. The authors should explore the source of this bias further. They correctly mention that the bias increases towards the higher wavenumbers where $H_2O$ is a stronger absorber. Does this suggest the 2006-2015 analogue profiles have systematically more WV than the 2005 references (albeit still within the threshold)? And that this could be leading to the systematic bias in the difference calculation? One can imagine that due to the small sample size, this could be possible.

Following the referee's previous comment, we will delete figure 1 and its specific analysis from the revised version of the paper. In the revised version, we will discuss in more detail the uncertainties related to results shown in figure 2. The new figure mentioned above will play an important role in this discussion.

Line 223-225: It is not clear why the authors are using just three $CO_2$ concentrations for three different years and then using a curve fit to identify the corresponding spectral radiances for years in between. Doesn't the Maona Loa data have $CO_2$ concentrations for all months and years within the studied timeframe? Some clarification would be helpful.

This analysis was performed for the data presented in figure 1 in the original version of the paper. Following the referee's previous comments, we will delete figure 1 and the discussion associated with it (including this part) in the revised version of the manuscript. We will mention the potential theoretical uncertainty due to $CO_2$ uncertainty while analysing figure 2, but we will not undertake the procedure mentioned above by the referee.

I view this work as an important proof of concept that AIRS is able to detect the influence of $CO_2$ on radiances in isolation. That alone, is worthy of publication. I wonder if this methodology can be applied longer-term to isolate and track trends in how $CO_2$ is influencing the climate (e.g. in the context of radiative forcing). Using analogues with similar T and WV would seem to be the only way to isolate $CO_2$ effects purely from observations, but radiative forcing itself is sensitive to the underlying climate state (e.g. Y. Huang et al. 2016). So on one hand, by trying to keep T and WV fixed, the method is not capturing the true direct effects of $CO_2$ on the climate. Additionally, one could imagine that, if this method is applied over a wider range of years, thus covering more climate change, it would become more difficult over time to find analogues within a reasonably small threshold and T and WV undergoes more changes. For the sake of appealing to

a broader audience, I encourage to authors to add some discussion along these lines, about the broader implications of their work. Maybe in their conclusion section.

These are critical aspects, and we agree with the comments, interpretation and suggestions of the referee. In the first version of the manuscript there is already some text in the conclusions along these lines. Specifically, we wrote 'In the future, variants of this methodology could be used to isolate the observational radiative impact of different physical and chemical properties of the climate system and as such provide a better observational depiction of the Earth's radiative forcing and of climate feedbacks.' In the revised version we will expand on the topic following the referee's suggestions, in particular on how to potentially generalize this methodology. The reference provided is an important one that we will add to the revised version of the paper.

Huang, Y., Tan, X., and Xia, Y. (2016), Inhomogeneous radiative forcing of homogeneous greenhouse gases, J. Geophys. Res. Atmos., 121, 2780–2789, doi:10.1002/2015JD024569.

---

## Author Comment (AC2)

**Reply to the review from referee 2**

Note: the original text from the review is in black and our replies are in blue.

This study used AIRS high spectral infrared radiance data to quantify the impact of increasing atmospheric CO2 concentration on the absorption of 15-um CO2 band after eliminating interference from water vapor and temperature changes. They found that the isolated signal from increased absorption by CO2 from AIRS is consistent with RT model simulations. Overall, this letter was well written. The message the authors trying to deliver is clear. The results from this study based on data driven analysis are important to confirm our understanding of CO2 greenhouse effect. However, given its short length, some important details are not there in the paper and therefore lead to confusions. I will list them below.

We would like to thank the referee for a detailed and constructive review. All the comments and suggestions by the referee have been addressed and have been extremely helpful in improving the manuscript. The revised version will be an improved manuscript because of the referee's comments and suggestions.

Some parameters in Experiments A and B are not the same. For example, you analyzed 2005-2015 for A but 2003 to 2012 for B. Is there a reason that the two experiments need to start from different years? Also, you explain the adoption of 1.2k and 1.2g/kg in the appendix for A, but in B, you used 1.4k and 1.4 g/kg. You need to justify these numbers.

Following the comments from referee 1, in the revised version of the paper we will not divide our study in experiments A and B. Rather we will focus on data obtained for a large number of reference profiles such as what was analysed and discussed in experiment B in the first version of the manuscript. Because of this, the current figure 1 (and its analysis) will be deleted from the revised version. So, the mismatch referred to above by the referee does not apply any longer in the analysis that will be shown in the revised version.

While there is some justification in the first version of the paper regarding the temperature and water vapor thresholds, in the revised version we will present a more detailed analysis of the data and of these thresholds. We will add a figure that compares the theoretically expected radiance differences due to CO2 changes versus radiance differences due to temperature and water vapor differences (constrained by the thresholds mentioned above) for the spectral region that we are focusing on. This will provide a sense of how large the radiance differences associated with these temperature and water vapor thresholds are expected to be, as compared to the CO2 differences.

In the experiments, you selected one set of profiles for A and 100 sets for B. It is not clear how the set of profiles were selected. How do you make sure they are representative of the temperature and h2o vertical distributions on Earth? It would be better if you can show some of the profiles in the appendix as well.

The reference profiles are selected in a random manner but subject to the following constraints: These profiles occur over the tropical/subtropical oceans (30 S to 30 N), in (almost) clear sky (cloud cover less than 10 %), during July of 2003 and with SSTs between 298 and 302 K. These

profiles are expected to be representative of the thermodynamic vertical structure of the (almost) clear-sky subtropical atmosphere. This regime is characterized by low values of subsidence and by conditionally unstable boundary layers populated by small amounts of shallow cumulus clouds or even completely clear boundary layers. This is a regime that occupies a large fraction of the Earth's surface and that plays a key climate role in the surface evaporation over the ocean and the outgoing longwave radiation. Following the referee's suggestion, in the revised version of the paper we will add to the appendix a figure illustrating the vertical structure of the reference profiles and a discussion on how representative and relevant they are.

"In experiment B, a key assumption is that the annual mean spectral radiance differences corresponding to each reference state are (to first order) not sensitive to the reference state itself for these selected reference profiles." You now have 100 sets of profiles and the corresponding spectral change. Can you use these results to justify your assumptions here? For example, are the temperature (or h2o) variabilities correlated with the spectral radiance differences?

This is already partly discussed in the first version of the paper when writing that, referring to figure 2, 'The theoretical annual mean differences are calculated based on the reference states. This allows to estimate not only the theoretical annual mean difference but also the associated standard deviation, which is shown as red shading. Note that the standard deviation is so small that it is almost imperceptible in the figure. This apparent lack of theoretical sensitivity to the reference states supports the key assumption, mentioned above'. But we will make this point clearer in the revised version of the paper.

In addition, and following some of the discussion above, in the revised version of the paper we will discuss in more detail the impact of the temperature and water vapor variability on the spectral radiance differences. This will be illustrated with a new figure as mentioned above.

In Figure 2, how large is the uncertainty for the observations? You have that for Figure 1 but not Figure 2. Also, the mismatch between observations and theoretical calculations are large over those $CO_2$ absorption line centers. The difference can be 0.04K for the lines on the left of 700 cm-1, which is larger than the expected spectra noise. You attributed this difference to $CO_2$ uncertainty. Can you reconcile the two if you increase the $CO_2$ in your RT model? It seems your current calculations have less absorption over those lines.

In the revised version of the paper, we will present a much more detailed discussion of the uncertainties associated with the results presented in figure 2, directly addressing the points raised by the referee. In addition, we will discuss in more detail the sensitivity of the final results to the scan angle and the filtering of potential outliers.

Minor comments:
Line 181-182, the absolute radiometric calibration accuracy is usually temperature dependence. For this 0.2K accuracy, is it relative to what temperature blackbody?

In the revised version of the paper, this issue will be clarified and more details regarding accuracy will be provided.

Line 206-207, do you have figure or reference to justify that the temperature profiles in AIRS/AMSU and AMSU MW-only are similar?

In the revised version of the paper a comparison between the AIRS/AMSU, the AMSU MW-only and the neural network (which is the first guess for the AIRS/AMSU retrieval) thermodynamic profiles will be added to the appendix.

In Longueville et al. (2021), the authors showed Figure 2 to illustrate the increased $CO_2$ absorption in the IASI spectra from 2008 to 2017, though they did not isolate it from the joint effects of temperature and h2o. This is a related reference for this study.
De Longueville, H., Clarisse, L., Whitburn, S., Franco, B., Bauduin, S., Clerbaux, C., et al. (2021). Identification of short and long-lived atmospheric trace gases from IASI space observations. Geophysical Research Letters, 48, e2020GL091742. https://doi. org/10.1029/2020GL091742

This is an important reference in the context of this study that will be mentioned and cited in the revised version of the paper.